# Surface Adsorption Properties and Layer Structures of Homogeneous Polyoxyethylene-Type Nonionic Surfactants in Quaternary-Ammonium-Salt-Type Amphiphilic Gemini Ionic Liquids with Oxygen- or Nitrogen-Containing Spacers

**DOI:** 10.3390/molecules25214881

**Published:** 2020-10-22

**Authors:** Risa Kawai, Maiko Niki, Shiho Yada, Tomokazu Yoshimura

**Affiliations:** Department of Chemistry, Faculty of Science, Nara Women’s University, Nara 630-8506, Japan; kawai-r@mse.suzuka-ct.ac.jp (R.K.); m9runner23@gmail.com (M.N.); yada@cc.nara-wu.ac.jp (S.Y.)

**Keywords:** quaternary ammonium salt, gemini ionic liquid, spacer, surface adsorption, layer structure, wide-angle X-ray scattering, viscosity

## Abstract

The amphiphilic ionic liquids containing an alkyl chain in molecules form nano-structure in the bulk, although they also show surface activity and form aggregates in aqueous solutions. Although insights into the layer structures of ionic liquids were obtained using X-ray and neutron scattering techniques, the nanostructures of ionic liquids remain unclear. Herein, the surface adsorption and bulk properties of homogeneous polyoxyethylene (EO)-type nonionic surfactants (C*_x_*EO_6_; *x* = 8, 12, or 16) were elucidated in quaternary-ammonium-salt-type amphiphilic gemini ionic liquids with oxygen or nitrogen-containing spacers [*2*C*_n_*(Spacer) NTf_2_; (Spacer) = (2-O-2), (2-O-2-O-2), (2-N-2), (2/2-N-2), (3), (5), or (6); *n* = 10, 12, or 14 for (2-O-2) and *n* = 12 for all other spacers] by surface tension, small- and wide-angle X-ray scattering, cryogenic transmission electron microscopy, and viscosity measurements. The surface tension of C_12_EO_6_ in 2C*_n_*(Spacer) NTf_2_ with oxygen-containing spacers increased with increasing concentration of C_12_EO_6_, becoming close to that of C_12_EO_6_ alone, indicating that the amphiphilic ionic liquid adsorbed at the interface was replaced with C*_x_*EO_6_. In contrast, both 2C*_n_*(Spacer) NTf_2_ with nitrogen-containing spacers and nonionic surfactants remained adsorbed at the interface at high concentrations. In the bulk, it was found that 2C*_n_*(Spacer) NTf_2_ formed layer structures, in which the spacing depended on the alkyl chain length of C*_x_*EO_6_. These insights are expected to advance the practical applications of amphiphilic ionic liquids such as ion permeation, drug solubilization, and energy delivery systems.

## 1. Introduction

Ionic liquids are salts consisting of only cations and anions that exhibit melting points below 100 °C. They have been the subject of extensive research because of their advantageous properties, such as low vapor pressure, high thermal stability, high conductivity, and unique solubility, and have attracted interest for use as novel and environment-friendly solvents [1,2,3,4,5]. Notably, the physicochemical properties of ionic liquids can be readily controlled by varying the combination of cations and anions. The ionic liquids containing a long alkyl chain in molecules show surface activities such as adsorption and aggregation, which are similar to classical surfactants. These ionic liquids are called amphiphilic ionic liquids [6].

In general, ionic liquids have non-uniform liquid structures consisting of nonpolar and polar domains. Recently, various studies have investigated the nanostructures of ionic liquids using small-angle X-ray scattering (SAXS), small-angle neutron scattering (SANS), and X-ray diffraction (XRD) [7,8,9,10,11,12,13,14]. Bradley et al. investigated the thermal behavior of ionic liquids based on 1-alkyl-3-methylimidazolium salts with long alkyl chains [C*_n_*mim X; *n* (alkyl chain length) = 12–18; X (counterion) = Cl^−^, Br^−^, bis(trifluoromethanesulfonyl)imide (NTf_2_^−^), or trifluoromethylsulfonate (OTf^−^)] [7]. These ionic liquids display layer structures, with one or more peaks in the low-angle region of the XRD pattern [7]. Moreover, C*_n_*mim X (X = Cl^−^ or PF_6_^−^) with *n* > 10 also form liquid-crystal phases [7]. Furthermore, non-uniform ionic liquid structures were reported by Padua et al. using molecular dynamics (MD) simulations [15,16] and by Triolo et al. using SAXS [10]. In the SAXS profiles of C*_n_*mim Cl, peaks in the *q* region of two to three nm^−1^ correspond to the spacing of the layer structure, which depends on the alkyl chain length [10]. Fujii et al. revealed the formation of a domain structure for C*_n_*mim NTf_2_ (*n* = 2, 4, 6, 8, 10, or 12) using SANS and MD simulations [11]. For C_8_mim X (X = Cl^−^, PF_6_^−^, or NTf_2_^−^), Kofu et al. investigated the effects of the counterion on the layer spacing using neutron diffraction (ND) and neutron spin echo spectroscopy [12]. Shimizu et al. observed three peaks in the *q* region of 1.6–20 nm^−1^, which were assigned to the layer spacing (2 nm^−1^), the cation–cation or anion–anion distances in polar networks (9 nm^−1^), and the cation–anion distance (13 nm^−1^) using MD simulations [13]. Bowlas et al. reported liquid-crystalline ionic liquids consisting of delocalized organic cations, such as imidazolium, pyridinium, and anions [14]. Despite these efforts, many features of the nanostructures of ionic liquids remain unclear because few investigations used X-ray and neutron scattering techniques.

Recently, we developed quaternary-ammonium-salt-type amphiphilic gemini ionic liquids by exchanging the counterions of the corresponding gemini surfactants, and revealed that the melting point can be lowered to ~40 °C by changing the degree of dissymmetry in the two alkyl chains [17]. The addition of nonionic surfactants to these amphiphilic gemini ionic liquids led to effective adsorption and orientation of both species at the interface [18]. We found that introducing different spacers containing oxygen or nitrogen lowered the melting point of quaternary-ammonium-salt-type amphiphilic gemini ionic liquids to below room temperature, and that the ionic liquids with nitrogen-containing spacers adsorbed efficiently at the air/water interface [19]. Further, we designed and synthesized quaternary-ammonium-salt-type amphiphilic trimeric ionic liquids with star and linear structures with melting points below zero degrees Celsius [20]. These trimeric ionic liquids exhibited excellent adsorption and orientation at the air/water interface compared with the corresponding amphiphilic monomeric ionic liquids. Moreover, they formed micelles at low concentrations, whereas the amphiphilic gemini ionic liquids did not. 

In this study, amphiphilic gemini ionic liquids with oxygen- or nitrogen-containing spacers [2C*_n_*(Spacer) NTf_2_; (Spacer) = (2-O-2), (2-O-2-O-2), (2-N-2), (2/2-N-2), (3), (5), or (6); *n* (alkyl chain length) = 10, 12, of 14 for (2-O-2) and *n* = 12 for all other spacers; Figure 1a] were used as a solvent for homogeneous polyoxyethylene (EO)-type nonionic surfactants [C*_x_*EO_6_; *x* (alkyl chain length) *x* = 8, 12, or 16; Figure 1b] to investigate the surface adsorption and bulk properties via surface tension, SAXS, wide-angle X-ray scattering (WAXS), cryogenic transmission electron microscopy (cryo-TEM), and viscosity measurements. In particular, the effects of the ionic liquid alkyl chain length, spacer length, and spacer structure, as well as the nonionic surfactant alkyl chain length, on the properties were determined. Furthermore, the temperature dependence of the layer structures of the amphiphilic gemini ionic liquids was investigated. Notably, the changes in surface tension upon addition of nonionic surfactants to the amphiphilic gemini ionic liquids depended on the spacer structure, as the gemini ionic liquids with oxygen-containing spacers were displaced from the interface, whereas those with nitrogen-containing spacers were not. Furthermore, layer structures were formed in the bulk, in which the spacing was affected by the spacer structure and length of the amphiphilic ionic liquid as well as the alkyl chain length of the nonionic surfactant.

## 2. Results and Discussion

### 2.1. Solubilities of Nonionic Surfactants in Quaternary-Ammonium-Salt-Type Amphiphilic Gemini Ionic Liquids

The phase-separation temperatures of the nonionic surfactants (C_8_EO_6_, C_12_EO_6_, and C_16_EO_6_) in amphiphilic gemini ionic liquids 2C*_n_*(Spacer) NTf_2_ were >100 °C. The cloud points of C_8_EO_6_, C_12_EO_6_, and C_16_EO_6_ in water were 72 [21], 52, and 32 °C [22], respectively, whereas those of C_12_EO_6_ in C_4_mimBF_4_ and C_4_mimPF_6_ were 60 and 119 °C, respectively [23]. As the phase-separation temperatures of the nonionic surfactants in the quaternary-ammonium-salt-type amphiphilic gemini ionic liquids were higher than their cloud points in water and imidazolium-based ionic liquids, the quaternary-ammonium-salt-type amphiphilic ionic liquids provided excellent solubility, even at high temperatures. This finding suggests that compatibility was provided by interactions between the alkyl chains of the amphiphilic ionic liquids and the alkyl chains of the surfactants. Notably, 2C_12_(2-O-2) NTf_2_ and 2C_12_(2-O-2-O-2) NTf_2_ did not show phase separation, despite having an EO chain in the spacer, indicating that the EO chain did not significantly affect the solubility of EO-type surfactants. The subsequent measurements of solutions using amphiphilic gemini ionic liquids were performed at 25, 30, or 40 °C. This was carried out because the solutions were clear at temperatures below their phase separation temperatures.

### 2.2. Surface Adsorption Behavior of Surfactants in Amphiphilic Gemini Ionic Liquids

Figure 2 shows the surface tension plots as a function of the volume fraction of C_12_EO_6_ in 2C*_n_*(Spacer) NTf_2_. The data at a volume fraction of 0 corresponded to the surface tension of the amphiphilic ionic liquids without added surfactant. The surface tensions of 2C*_n_*(Spacer) NTf_2_ were 29.6–31.1 mN m^−1^, indicating that the spacer structure and alkyl chain length did not affect the surface tension. However, upon addition of C_12_EO_6_, the surface tension plots differed according to the spacer structure of the amphiphilic ionic liquids. For 2C*_n_*(2-O-2) NTf_2_ (*n* = 10, 12, or 14), the surface tension increased with increasing volume fraction of C_12_EO_6_, until it was similar to that of C_12_EO_6_ (35.2 mN m^−1^). This observation suggests that 2C*_n_*(2-O-2) NTf_2_ adsorbed at the interface was replaced with C_12_EO_6_. This behavior was similar to that observed for C_12_EO_6_ in the amphiphilic monomeric ionic liquid C_4_ NTf_2_ [17]. In contrast, for 2C_12_(2-N-2) NTf_2_ and 2C_12_(2/2-N-2) NTf_2_, the surface tension only increased slightly with increasing volume fraction of C_12_EO_6_, reaching a constant value of ~32 mN m^−1^. Thus, it is considered that the ionic liquids remained at the interface and were not entirely replaced with C_12_EO_6_. This behavior suggests that the methyl group and methylene chain in the spacers of 2C_12_(2-N-2) NTf_2_ and 2C_12_(2/2-N-2) NTf_2_ were adsorbed efficiently at the interface.

### 2.3. Layer Structure of Amphiphilic Gemini Ionic Liquids

Figure 3 shows the WAXS profiles of 2C*_n_*(2-O-2) NTf_2_ (*n* = 4, 6, 8, 10, 12, or 14), where *q* is the scattering vector and *I*(*q*) is the scattering intensity. Three peaks were observed for 2C*_n_*(2-O-2) NTf_2_ with *n* = 6–14, indicating the formation of a layer structure. Using the relation *d* = 2*π*/*q*, peaks *q*_1_ (2–3 nm^−1^), *q*_2_ (~8 nm^−1^), and *q*_3_ (~13 nm^−1^) give *d*_1_, *d*_2_, and *d*_3_ values corresponding to the layer spacing, the cation–cation or anion–anion distances in the polar network, and the cation–anion distance, respectively [9,10,11,14]. For 2C*_n_*(2-O-2) NTf_2_ with *n* = 6, 8, 10, 12, and 14, the *d*_1_ values increased with increasing alkyl chain length (1.6, 2.0, 2.4, 2.8, and 3.0 nm, respectively). The WAXS profiles of amphiphilic monomeric ionic liquids C*_n_* NTf_2_ (*n* = 4, 6, 8, 10, or 12) with alkyl chains longer than 6 also showed three peaks (Appendix A), indicating the formation of a layer structure similar to that of the amphiphilic gemini ionic liquids. The *d*_1_ value of the layer structure of *C_n_* NTf_2_ also increased with increasing alkyl chain length. However, the *d*_1_ values of 2C*_n_*(2-O-2) NTf_2_ were larger than those of C*_n_* NTf_2_ at all alkyl chain lengths (Appendix A) because of the bulky (2-O-2) spacer. The critical lengths of the alkyl chains for *n* = 6, 8, 10, 12, and 14 were determined to be 1.8, 2.3, 2.8, 3.3, and 3.9 nm, respectively, using the Tanford equation [24]. However, the *d*_1_ values of 2C*_n_*(2-O-2) NTf_2_ were less than twice the critical lengths of the alkyl chains, which suggests that the alkyl chains of the amphiphilic gemini ionic liquids were readily interlocked. 

Figure 4 shows the WAXS profiles of amphiphilic gemini ionic liquids 2C_12_(Spacer) NTf_2_ with various spacers. Three peaks were observed in the profiles, indicating that all these amphiphilic ionic liquids form layer structures. Table 1 shows the *d*_1_, *d*_2_, and *d*_3_ values calculated from the *q*_1_, *q*_2_, and *q*_3_ peaks of 2C_12_(Spacer) NTf_2_. The *d*_1_ value of the layer structure formed by 2C_12_(Spacer) NTf_2_ depended slightly on the spacer structure, whereas *d*_2_ and *d*_3_ were not influenced by the spacer structure. This suggests that the *d*_2_ corresponds to the distance between faced cations instead of that between adjacent cations. Layer spacing *d*_1_ increased in the following order, based on the spacer present: (5) = (2-O-2-O-2) < (2-N-2) = (2/2-N-2) ≅ (2-O-2). The *d*_1_ values of 2C_12_(Spacer) NTf_2_ with nitrogen- and oxygen-containing spacers were larger than that of 2C_12_-5 NTf_2_ with a pentylene chain in the spacer, likely because the spacers containing nitrogen and oxygen were more rigid. The *d*_1_ values of 2C_12_-*s* NTf_2_ increased as the spacer chain length increased from 3 to 5 and then 6, suggesting that the flexible spacer faced the alkyl chain side and that the alkyl chains are not readily interlocked for longer spacers. In contrast, the *d*_1_ value of 2C_12_(2-O-2-O-2) NTf_2_ was the same as that of 2C_12_-5 NTf_2_, despite the (2-O-2-O-2) spacer being longer than the pentylene chain. It is considered that the spacer containing the dioxyethylene chain cannot easily bend toward the alkyl chain side because the affinity between the hydrophilic dioxyethylene chains and hydrophobic alkyl chains is low, and thus the alkyl chains of 2C_12_(2-O-2-O-2) NTf_2_ can readily interlock. Furthermore, cryo-TEM imaging of 2C_12_(2-O-2) NTf_2_ (Figure 5) revealed the formation of a layer structure with a layer spacing of 3–5 nm, which corresponded to the value calculated from the WAXS profile. The controversy about nanostructures of ionic liquids obtained from X-ray and neutron scattering techniques has continued; however, it is noteworthy that the layer structure of ionic liquid could be directly visualized using cryo-TEM. 

The effects of temperature on the layer structures of amphiphilic gemini ionic liquids 2C_12_(Spacer) NTf_2_ and amphiphilic monomeric ionic liquid C_12_ NTf_2_ were investigated. In all cases, as the temperature increased, the layer spacing (*d*_1_) decreased, the cation–cation or anion–anion distance in the polar network (*d*_2_) was almost unchanged, and the cation–anion distance (*d*_3_) increased (Appendix A). This behavior suggests that the alkyl chains of amphiphilic ionic liquids become more flexible at higher temperatures, which facilitates interlocking in the bilayer structure. The increase in *d*_3_ with increasing temperature is thought to be due to an increase in cation mobility at higher temperatures. 

### 2.4. Layer Structure of Amphiphilic Gemini Ionic Liquids and Nonionic Surfactants

The effects of EO-type nonionic surfactant C_12_EO_6_ on the layer structure of amphiphilic gemini ionic liquids 2C_12_(Spacer) NTf_2_ were investigated. Figure 6 shows the WAXS profiles of 0–1000 mmol dm^−3^ C_12_EO_6_ in 2C_12_(2-O-2) NTf_2_ at 25 °C. Similar to the profiles of 2C*_n_*(Spacer) NTf_2_ alone, three peaks were observed in the profiles of 2C_12_(2-O-2) NTf_2_ at all C_12_EO_6_ concentrations, indicating the formation of layer structures. Peaks *q*_1_ (2–3 nm^−1^), *q*_2_ (8 nm^−1^), and *q*_3_ (13 nm^−1^) corresponded to the layer spacing (*d*_1_), the distance between faced cations in layers (*d*_2_), and the cation–counterion distance (*d*_3_), respectively. Peak *q*_1_ shifted to the low*-q* region as the concentration of C_12_EO_6_ increased from 10 to 1000 mmol dm^−3^, indicating a decrease in *d*_1_. Thus, it is considered that C_12_EO_6_ is interlocked within the layer structure of 2C_12_(2-O-2) NTf_2_. Peak *q*_2_ at approximately 8 nm^−1^ became broader as the concentration of C_12_EO_6_ increased and disappeared at 1000 mmol dm^−3^. This behavior indicated that the order of the hydrophilic moieties in the amphiphilic ionic liquids decreased with increasing concentration of C_12_EO_6_. This change occured because the EO chain of the surfactant was larger than the ammonium group of the amphiphilic ionic liquid, and the layer was formed by bending the EO chain. Peak *q*_3_ did not change significantly with increasing concentration of C_12_EO_6_. Similar behavior was observed for 2C_12_(Spacer) NTf_2_ with other spacers (Appendix A).

Figure 7 shows the *d*_1_ values of C*_x_*EO_6_ (*x* = 8, 12, or 16) in 2C*_n_*(2-O-2) NTf_2_ (*n* = 10, 12, or 14) as a function of the volume fraction of C*_x_*EO_6_. Layer spacing *d*_1_, corresponding to peak *q*_1_ at 2–3 nm^−1^ in the SAXS profile, varied according to the alkyl chain lengths of the amphiphilic ionic liquids and surfactants. In the case of C_8_EO_6_ in 2C_12_(2-O-2) NTf_2_, *d*_1_ decreased slightly with increasing C_8_EO_6_ concentration, whereas the opposite trend was observed for C_12_EO_6_ and C_16_EO_6_ in 2C_12_(2-O-2) NTf_2_. The *d*_1_ values for 1000 mmol dm^−3^ C*_x_*EO_6_ (*x* = 8, 12, or 16) in 2C_12_(2-O-2) NTf_2_ were 2.6, 3.0, and 3.3 nm, respectively. The *d*_1_ values of C*_x_*EO_6_ (*x* = 8, 12, or 16) alone were 2.6, 3.1, and 3.3 nm, respectively, and as the concentration of C*_x_*EO_6_ increased, the spacing of the layer structure consisting of 2C_12_(2-O-2) NTf_2_ and C*_x_*EO_6_ became more similar to that of C*_x_*EO_6_ alone. The *d*_1_ values of 1000 mmol dm^−3^ C*_x_*EO_6_ (*x* = 8, 12, or 16) in 2C_10_(2-O-2) NTf_2_ and 2C_14_(2-O-2) NTf_2_ increased with increasing alkyl chain length of C*_x_*EO_6_ (Appendix A). The critical lengths of the alkyl chains of *n* = 8, 12, and 16 were determined to be 1.2, 1.7, and 2.2 nm, respectively, using the Tanford equation [24]. For C_8_EO_6_, the *d*_1_ value was more than two times greater than the alkyl chain length, whereas for C_16_EO_6_, the *d*_1_ value was less than two times greater than the alkyl chain of length. The *d*_1_ value of 1000 mmol dm^−3^ C_8_EO_6_ in 2C*_n_*(2-O-2) NTf_2_ increased from 2.5 to 2.7 nm as the alkyl chain length of 2C*_n_*(2-O-2) NTf_2_ increased from *n =* 10 to *n =* 14, whereas for C_16_EO_6_, the *d*_1_ value remained constant (3.3 nm). Different behaviors were also observed depending on the alkyl chain length of the ionic liquid (Appendix A). Overall, the spacing of the layer structure consisting of amphiphilic ionic liquids a nd nonionic surfactants depends more on the alkyl chain length of the nonionic surfactant than on the alkyl chain length of the ionic liquid.

Figure 8 shows the relationship between the zero-shear viscosity (*η*_0_) and the concentration of C*_x_*EO_6_ (*x* = 12 or 16) in 2C*_n_*(Spacer) NTf_2_. The *η*_0_ value gradually decreased as the concentration of C*_x_*EO_6_ increased up to 100 mmol dm^−3^, and then significantly decreased at higher concentrations. This behavior suggests that the nonionic surfactants were interlocked within the layer structure formed by the amphiphilic ionic liquids and that the intermolecular interactions between of ionic liquids were weakened. The cryo-TEM image of 750 mmol dm^−3^ C_12_EO_6_ in 2C_12_(2-O-2) NTf_2_ showed a striped structure (Appendix A), while it was ambiguous compared to 2C_12_(2-O-2) NTf_2_ alone (Figure 5). As described above, peak *q*_1_ in the WAXS profiles of the amphiphilic liquids broadened with increasing concentration of C_12_EO_6_, indicating that the order of the layer structure was decreased by the addition of the nonionic surfactant to the amphiphilic ionic liquids. Thus, the formation of a layer structure consisting of both the amphiphilic ionic liquids and nonionic surfactants was revealed.

## 3. Materials and Methods

### 3.1. Materials

The quaternary-ammonium-salt-type amphiphilic gemini ionic liquids, bis[2-(*N*-alkyl-*N*,*N*- dimethylammonio)ethyl]ether bis(trifluoromethanesulfonyl)amide (2C*_n_*(2-O-2) NTf_2_, *n* = 4, 6, 8, 10, 12, or 14), 1,2-bis[2-(*N*-dodecyl-*N*,*N*-dimethylammonio)ethoxy]ethane bis(trifluoromethanesulfo- nyl)amide (2C_12_(2-O-2-O-2) NTf_2_), bis(*N*-dodecyl-*N*,*N*-dimethylethylammonium)-*N’*-methylamine bis(trifluoromethanesulfonyl)amide (2C_12_(2-N-2) NTf_2_), 1-[2-(*N*-dodecyl-*N*,*N*-dimethylammonio- ethyl)]-4-dodecylmethylpiperazinium bis(trifluoromethanesulfonyl)amide (2C_12_(2/2-N-2) NTf_2_), propanediyl-1,5-bis(dimethylalkylammonium) bis(trifluoromethanesulfonyl) amide (2C_12_-3 NTf_2_), pentanediyl-1,5-bis(dimethylalkylammonium) bis(trifluoromethanesulfonyl) amide (2C_12_-5 NTf_2_), and hexanediyl-1,6-bis(dimethylalkylammonium) bis(trifluoromethanesulfonyl)amide (2C_12_-6 NTf_2_) were synthesized according to our previous report [19]. The melting points of 2C_10_(2-O-2) NTf_2_, 2C_12_(2-O-2) NTf_2_, 2C_14_(2-O-2) NTf_2_, 2C_12_(2-O-2-O-2) NTf_2_, 2C_12_(2-N-2) NTf_2_, 2C_12_(2/2-N-2) NTf_2_, 2C_12_-5 NTf_2_, and 2C_12_-6 NTf_2_ were −1.0, −8.3, 34.6, −16.4, −5.3, 20.3, −10.3, and −12.8 °C, respectively [19]. The homogeneous EO-type nonionic surfactants hexaoxyethylene dodecyl ether (C_12_EO_6_) and hexaoxyethylene hexadecyl ether (C_16_EO_6_) were supplied by Nikko Chemicals Co., Ltd. (Tokyo, Japan) and were used as received. Hexaoxyethylene octyl ether (C_8_EO_6_) was obtained from the reaction of triethylene glycol with trioxyethylene octyl ether, which was obtained by reacting octyl bromide and triethylene glycol, and was purified by column chromatography (30 mm inner diameter, 100 mL silica gel (Wakosil ^®^C-200)) using a mixture of ethyl acetate and methanol (9:1, *v/v*).

### 3.2. Phase-Separation Temperature

Clear solutions were prepared by dissolving 1.0 wt% nonionic surfactant (C_8_EO_6_, C_12_EO_6_, or C_16_EO_6_) in a hot ionic liquid and then placing the solution in a refrigerator at ~5 °C for at least 24 h. The phase-separation temperature was determined by gradually increasing the temperature of the solution from 5 to 100 °C. In cases where the solution did not separate into two phases, the phase separation temperature was determined to be >100 °C. 

### 3.3. Surface Tension

The surface tensions of the solutions consisting of surfactants in amphiphilic gemini ionic liquids were measured using a Teclis Tracker tensiometer (Lyon, France) using the pendant drop technique.

### 3.4. SAXS and WAXS

SAXS and WAXS measurements were conducted using a SAXS instrument installed at the BL40B2 beamline in SPring-8 (Hyogo, Japan). The X-ray wavelength was 0.7 Å, the sample-to-detector distance was 2.0 m for SAXS and 0.423 m for WAXS, and a large-area pixel detector (PILATUS-3S 2M, DECTRIS Ltd., Baden, Switzerland) was used. The two-dimensional SAXS and WAXS images obtained with Pilatus were converted into one-dimensional scattering intensity versus *q* profiles by circular averaging. To obtain the scattering intensity (*I*(*q*)) at each *q* value, background scattering was subtracted from the raw scattering data after an appropriate transmittance correction. The *q* range was 0.1–5 nm^−1^ for SAXS and 1–35 nm^−1^ for WAXS [*q* = (4*π*/*λ*)sin(*θ*/2), where *λ* and *θ* represent the wavelength and scattering angle, respectively]. The exposure time for each sample was 20 s. 

### 3.5. Viscosity

Viscosities were measured using a Brookfield DV-2T system (Middleborough, MA, USA). The zero-shear viscosities of the solutions consisting of surfactants in amphiphilic gemini ionic liquids were determined from the relationship between the viscosity and shear rate, as measured by increasing the rotational speed of the spindle.

## 4. Conclusions

In this study, the surface adsorption and bulk properties of EO-type nonionic surfactants in quaternary-ammonium-salt-type amphiphilic gemini ionic liquids with oxygen- or nitrogen-containing spacers were investigated, with a focus on the effects of the alkyl chain length, spacer structure, and spacer length of the amphiphilic ionic liquid and the alkyl chain length of the surfactant. The addition of the nonionic surfactants to the ionic liquids increased the surface tension, and the observed behavior differed depending on the spacer structure. When surfactants were added, gemini ionic liquids with oxygen-containing spacers adsorbed at the interface were replaced with nonionic surfactants, whereas those with nitrogen-containing spacers remained at the interface and were not entirely replaced with nonionic surfactants. Furthermore, the amphiphilic gemini ionic liquids with alkyl chains longer than six formed layer structures in the bulk, similar to those formed by monomeric amphiphilic ionic liquids. Notably, the spacing of the layer structures formed by the amphiphilic gemini ionic liquids was affected by the spacer structure and length, as revealed by SAXS and WAXS. These results were supporting by cryo-TEM imaging, which allowed the layer structure formed by the amphiphilic gemini ionic liquids in the bulk to be directly visualized for the first time. The visualization using cryo-TEM will be useful for investigating nanostructure of ionic liquids in the future. The addition of nonionic surfactants to the amphiphilic ionic liquids led to a decrease in the order of the layers. In addition, change in the layer spacing with an increase in the nonionic surfactant concentration showed a greater dependence on the alkyl chain length of the nonionic surfactants than on that of the ionic liquid. Interestingly, these amphiphilic gemini ionic liquids form unique nanostructures, and the bulk layer structure can be controlled by changing the spacer structure of the amphiphilic ionic liquid and the alkyl chain length of the added nonionic surfactant. These bulk properties can be used for solubilization of various compounds as well as surfactants in ionic liquids. We expect the findings to aid in the future development of amphiphilic ionic liquids for use in ion permeation, drug solubilization, information and energy delivery systems, and various other industrial applications. 

## Figures and Tables

**Figure 1 molecules-25-04881-f001:**
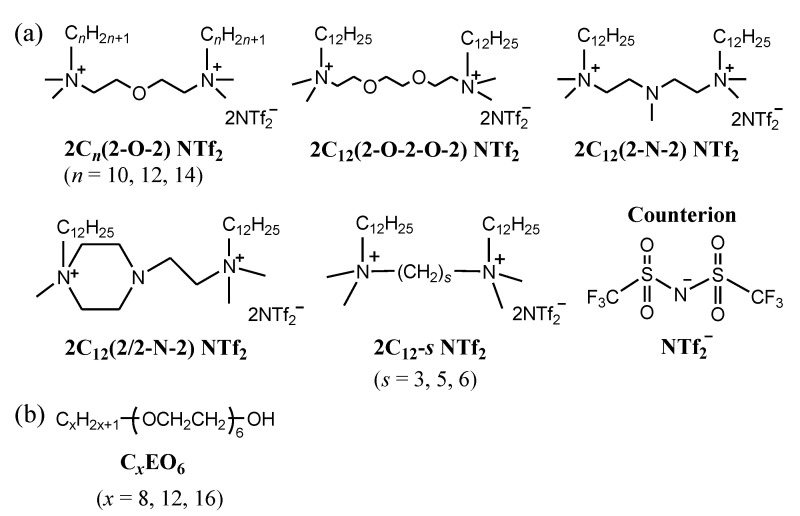
Chemical structures of (**a**) quaternary-ammonium-salt-type amphiphilic gemini ionic liquids and (**b**) nonionic surfactants.

**Figure 2 molecules-25-04881-f002:**
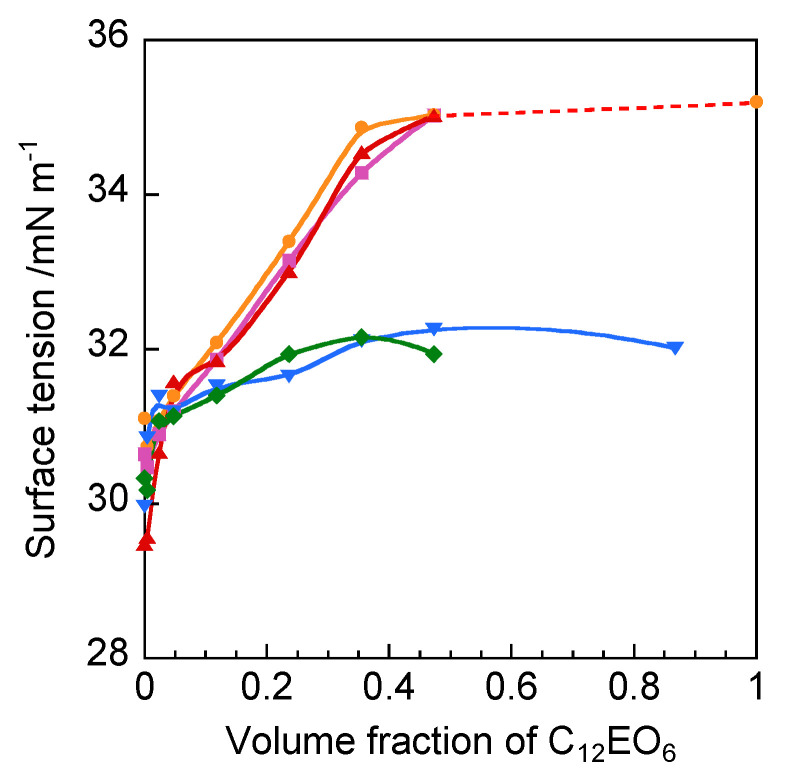
Surface tension as a function of the volume fraction of C_12_EO_6_ in amphiphilic gemini ionic liquids at 25 °C: 2C_10_(2-O-2) NTf_2_ (orange ●), 2C_12_(2-O-2) NTf_2_ (pink ■), 2C_14_(2-O-2) NTf_2_ (40 °C, red ▲), 2C_12_(2-N-2) NTf_2_ (blue ▼), and 2C_12_(2/2-N-2) NTf_2_ (30 °C, green ◆).

**Figure 3 molecules-25-04881-f003:**
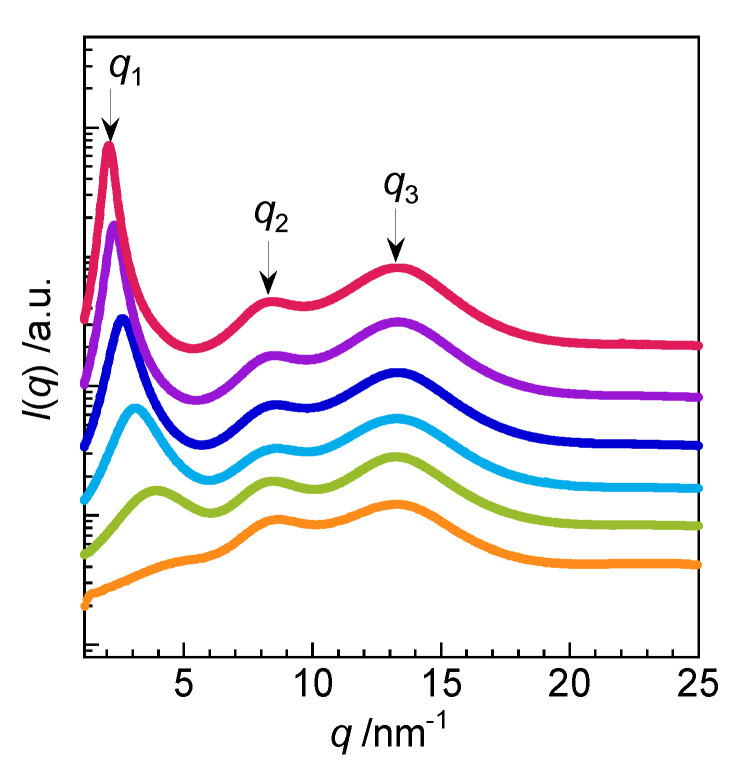
WAXS profiles of 2C*_n_*(2-O-2) NTf_2_ at 25 °C: *n* = 4 (orange ●), *n* = 6 (yellow green ●), *n* = 8 (sky blue ●), *n* = 10 (blue ●), *n* = 12 (purple ●), and *n* = 14 (40 °C, red ●).

**Figure 4 molecules-25-04881-f004:**
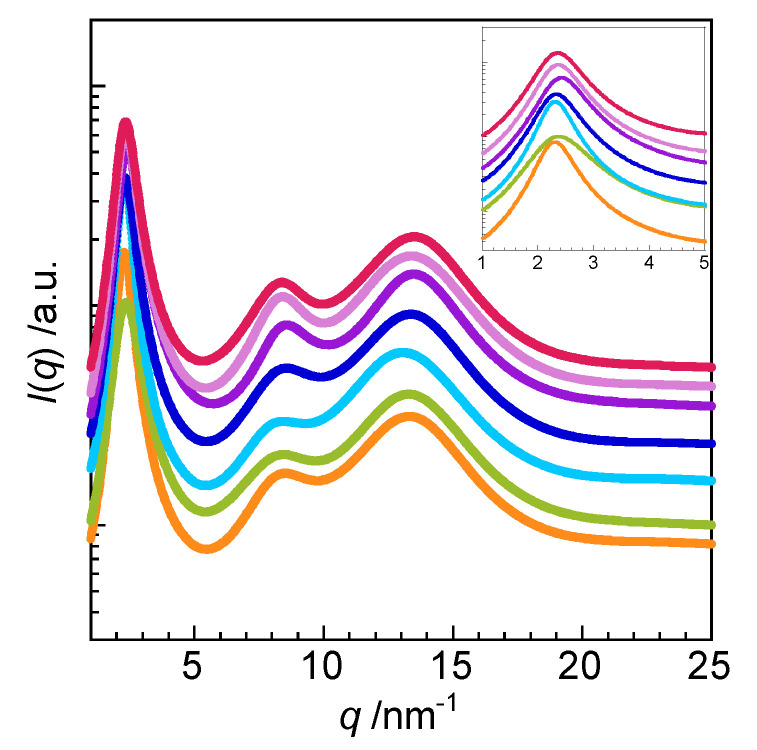
WAXS profiles of 2C_12_(Spacer) NTf_2_ at 25 °C: (2-O-2) (orange ●), (2-O-2-O-2) (yellow green ●), (2-N-2) (sky blue ●), (2/2-N-2) (30 °C, blue ●), (3) (purple ●), (5) (pink ●), and (6) (red ●).

**Figure 5 molecules-25-04881-f005:**
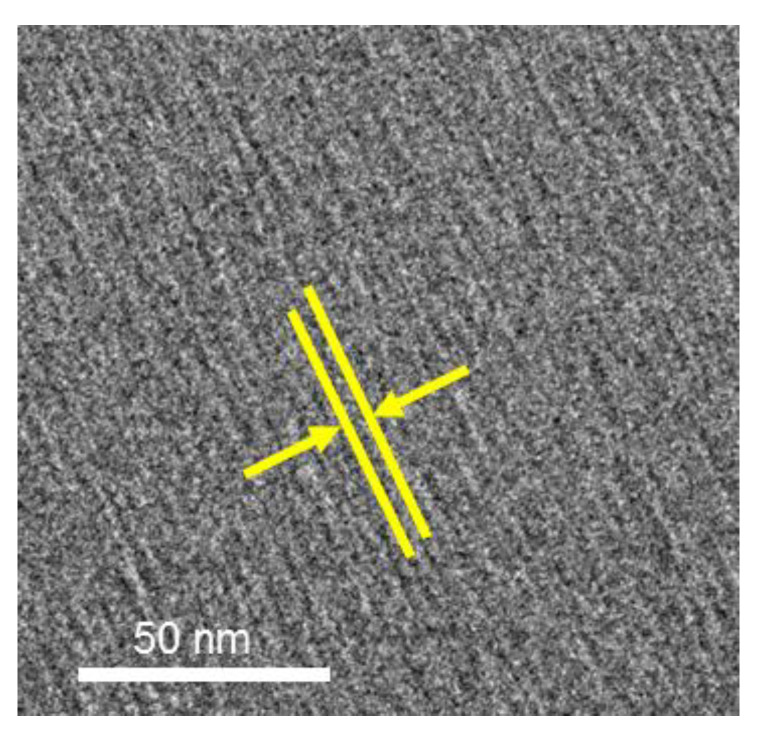
Cryo-TEM image of 2C_12_(2-O-2) NTf_2_.

**Figure 6 molecules-25-04881-f006:**
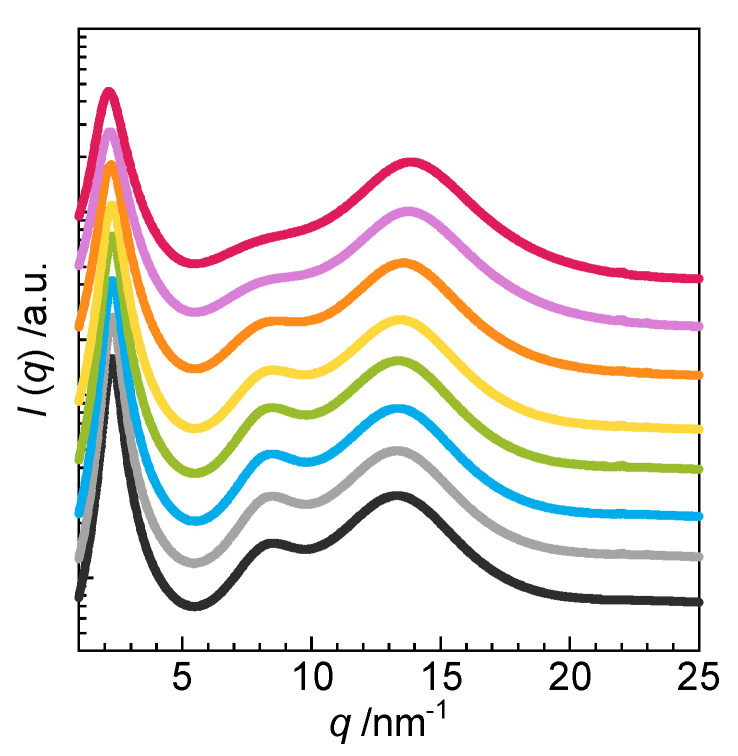
WAXS profiles of nonionic surfactant C_12_EO_6_ in amphiphilic gemini ionic liquid 2C_12_(2-O-2) NTf_2_ at 25 °C: 0 (black ●), 10 (gray ●), 50 (sky blue ●), 100 (yellow green ●), 250 (yellow ●), 500 (orange ●), 750 (pink ●), and 1000 mmol dm^−3^ (red ●).

**Figure 7 molecules-25-04881-f007:**
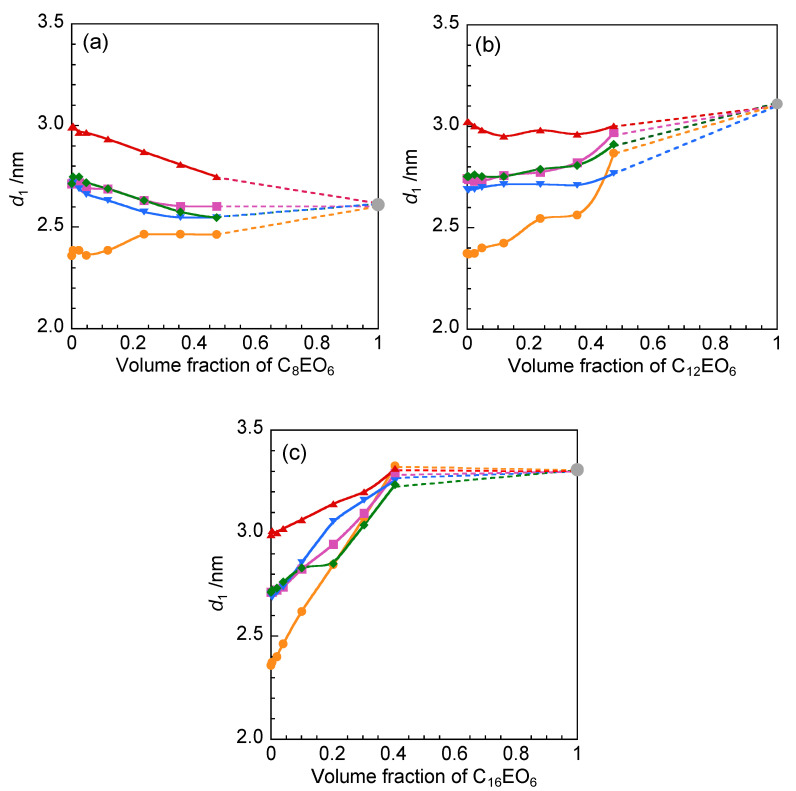
Variation in layer spacing *d*_1_ with the volume fraction of (**a**) C_8_EO_6_, (**b**) C_12_EO_6_, and (**c**) C_16_EO_6_ in 2C*_n_*(Spacer) NTf_2_ at 25 °C: 2C_10_(2-O-2) NTf_2_ (orange ●), 2C_12_(2-O-2) NTf_2_ (pink ■), 2C_14_(2-O-2) NTf_2_ (40 °C, red ▲), 2C_12_(2-N-2) NTf_2_ (blue ▼), and 2C_12_(2/2-N-2) NTf_2_ (30 °C, green ◆).

**Figure 8 molecules-25-04881-f008:**
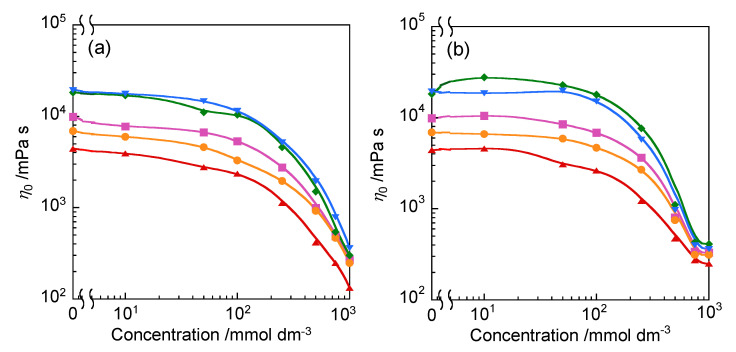
Variation in the zero-shear viscosity (*η*_0_) with the concentration of (**a**) C_12_EO_6_ and (**b**) C_16_EO_6_ in 2C*_n_*(Spacer) NTf_2_ at 25 °C: 2C_10_(2-*O*-2) NTf_2_ (orange ●), 2C_12_(2-O-2) NTf_2_ (pink ■), 2C_14_(2-O-2) NTf_2_ (40 °C, red ▲), 2C_12_(2-N-2) NTf_2_ (blue ▼), and 2C_12_(2/2–N–2) NTf_2_ (30 °C, green ◆).

**Table 1 molecules-25-04881-t001:** *d*_1_, *d*_2_, and *d*_3_ values obtained from the WAXS profiles of amphiphilic gemini ionic liquids 2C_12_(Spacer) NTf_2_.

Spacer	*d*_1_/nm	*d*_2_/nm	*d*_3_/nm
(2-O-2)	2.74	0.74	0.47
(2-O-2-O-2)	2.65	0.74	0.47
(2-N-2)	2.72	0.73	0.47
(2/2-N-2)	2.72	0.74	0.48
(3)	2.59	0.73	0.47
(5)	2.65	0.75	0.47
(6)	2.67	0.75	0.46

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
