# Peer review of "Surface Adsorption Properties and Layer Structures of Homogeneous Polyoxyethylene-Type Nonionic Surfactants in Quaternary-Ammonium-Salt-Type Amphiphilic Gemini Ionic Liquids with Oxygen- or Nitrogen-Containing Spacers"

_molecules, 2020, doi:10.3390/molecules25214881_

Round 1
Reviewer 1 Report
The authors report on a serious of ionic liquids with adapted organic spacers. They have studied the wetting behaviour and the internal layer structure by x-ray scattering techniques. Overall, the manuscript is well written with a slight tendency to resemble a technical report. The motivation and scientific discussion is rather brief but this seems to be a matter of writing style and personal preference. The topic is interesting to the readership of Molecules and the presentation quality is not particularly good but still appropriate. I would suggest that the authors put more emphasis on the significance of their findings instead of focussing on the data itself. Nevertheless, I am convinced that the manuscript will reach the right target audience in this journal. From a technical perspective, there is nothing particular to critique or to improve. The main weakness of this contribution is its low significance - or at least the way its significance is described in the text. However, another issues comes to mind, which is that the title seems a bit erroneous: "surface adsorption properties" is not correct, I suggest to use "surface wetting properties", instead. Finally, I recommend acceptance after minor text revisions, as described above.
Author Response
We would like to thank the editor and reviewers for their useful comments. We found these comments most helpful and their application has significantly improved our paper. As indicated below, we have checked all the general and specific comments provided by the Referees and have made necessary changes according to their indications.
Reviewer: 1
Comments:
The authors report on a serious of ionic liquids with adapted organic spacers. They have studied the wetting behavior and the internal layer structure by x-ray scattering techniques. Overall, the manuscript is well written with a slight tendency to resemble a technical report. The motivation and scientific discussion is rather brief but this seems to be interesting to the readership of Molecules and the presentation quality is not particularly good but still appropriate. I would suggest that the authors put more emphasis on the significance of their findings instead of focusing on the data itself. Nevertheless, I am convinced that the manuscript will reach the right target audience in this journal. From a technical perspective, there is nothing particular to critique or to improve. The main weakness of this contribution is its low significance – or at least the way its significance is described in the text. However, another issues comes to mind, which is that the title seems a bit erroneous: “surface adsorption properties” is not correct, I suggest to use “surface wetting properties,” instead. Finally, I recommend acceptance after minor text revisions, as described above.
Response to comments from Reviewer 1:
We would like to thank Reviewer 1 for the useful comments provided. We have answered the reviewer’s comments below and have made necessary changes according to their indications.
We have added the sentences “The controversy about nanostructures of ionic liquids obtained from X-ray and neutron scattering techniques has continued, however, it is noteworthy that layer structure of ionic liquid could be directly visualized using cryo-TEM.” about significance of the results on p. 5, lines 184−187.
It seems that “surface adsorption properties” in title would not be problematic, because “surface adsorption” in this study does not imitate the phenomenon on solid surface but that at the air/ionic liquid interface. In general, wettability is evaluated by contact angle of droplets on solid surface. Therefore, we think this title does not need to be changed.
Reviewer 2 Report
This article (Manuscript number molecules-968493-peer-review-v1) presents information about the surface adsorption and bulk properties of EO-type nonionic surfactants in quaternary-ammonium-salt-type amphiphilic gemini ionic liquids with oxygen- or nitrogen-containing spacers. The authors focus mainly on the effects of the alkyl chain length, spacer structure, and spacer length of the amphiphilic ionic liquid and the alkyl chain length of the surfactant. Therefore, the resulting phenomena of amphiphilic ionic liquids are very promising in various industrial applications, such as ion permeation, drug solubilization, and energy delivery systems.
On the whole, the overall originality of the concept used here is medium-high. Nevertheless, I would recommend publication of this article in Molecules on the condition a major revision of the manuscript will be carried out and the following points will be taken into consideration.
Detailed comments:
- The abstract needs to be well written with future prospects of the work and describe in short the concept of amphiphilic ionic liquids phenomena.
- There is a lack of important references related to the e.g. methods used for receiving well-defined amphiphilic ionic liquids. Furthermore, the introduction should be worked out - so as to show the full state of knowledge on this topic.
- More detailed results discussion should be provided. The discussion section appears to be a collection of data, however, the author’s self-opinion is of importance while drafting a results section of this type.
- The conclusion reflects an overall summary of the field with further extension and includes future prospective - I would suggest clarifying this section.
- Authors should work on editing text, spacing pause, Justify, fonts, italics, etc.
- References should be adapted to the requirements of the journal.
- The style and grammar leave much to be desired in many places, some parts of the text are difficult to understand.
After completing the above-mentioned corrections this work will be more readable. Therefore, it will be useful for the readers of Molecules.
Author Response
We would like to thank the editor and reviewers for their useful comments. We found these comments most helpful and their application has significantly improved our paper. As indicated below, we have checked all the general and specific comments provided by the Referees and have made necessary changes according to their indications.
Reviewer: 2
Comments:
This article (Manuscript number molecules-968493-peerreview-v1) presents information about the surface adsorption and bulk properties of EO-type nonionic surfactants in quaternary-ammonium-salt-type amphiphilic gemini ionic liquids with oxygen- or nitrogen-containing spacers. The authors focus mainly on the effects of the alkyl chain length, spacer structure, and spacer length of the amphiphilic ionic liquid and the alkyl chain length of the surfactant. Therefore, the resulting phenomena of amphiphilic ionic liquids are very promising in various industrial applications, such as ion permeation, drug solubilization, and energy delivery systems.
Responses to the comments from Reviewer 2:
We would like to thank Reviewer 2 for the useful comments provided. We have answered the reviewer’s comments below and have made the necessary changes according to their indications.
Comment:
(1) The abstract needs to be well written with future prospects of the work and describe in short the concept of amphiphilic ionic liquids phenomena.
Response:
- Thank you for your comments. We have added the sentences “such as ion permeation, drug solubilization, and energy delivery systems” about future prospects on p.1, lines 28. Further, we have added the sentences “The amphiphilic ionic liquids containing an alkyl chain in molecule form nano-structure in the bulk, although they also show surface activity and form aggregates in aqueous solutions” about the phenomena of amphiphilic ionic liquids on p. 1, lines 12–13.
Comment:
(2) There is a lack of important references related to the e.g. methods used for receiving well-defined amphiphilic ionic liquids. Furthermore, the introduction should be worked out – so as to show the full state of knowledge on this topic.
Response:
- Thank you for pointing that out. We have added the sentences “The ionic liquids containing long alkyl chain in molecule show surface activities such as adsorption and aggregation, which are similar to classical surfactants. These ionic liquids are called amphiphilic ionic liquids [6].” on p.1, lines 38−40. Furthermore, we have added the reference (6. Chamiot, B.; Rizzi, C.; Gaillon, L.; Sirieix-Plénet, J.; Lelièvre J. Redox-switched amphiphilic ionic liquid behavior in aqueous solution. Langmuir 2009, 25, 1311−1315.) on 10, lines 392−393, and amended the other references number.
Comment:
(3) More detailed results discussion should be provided. The discussion section appears to be a collection of data, however, the author’s self-opinion is of importance while drafting and results section of this type.
Response:
- Thank you for your comments. We have discussed layer structure as described on p. 4, lines 151−153, “the d1 values of 2Cn(2-O-2) NTf2 were less than twice the critical lengths of the alkyl chains, which suggests that the alkyl chains of the amphiphilic gemini ionic liquids were readily interlocked.”.
We have discussed temperature dependence as described on p.6, lines 221−223, “This behavior suggests that the alkyl chains of amphiphilic ionic liquids become more flexible at higher temperatures, which facilitates interlocking in the bilayer structure. The increase in d3 with increasing temperature is thought to be due to an increase in cation mobility at higher temperatures.”
Further, we have added the sentences “This suggests that the d2 corresponded to the distance between faced cations instead of that between adjacent cations.” on p.5, lines 170−172.
We have discussed the effects of addition of EO-type nonionic surfactant to the amphiphilic gemini ionic liquids on p.6, lines 234−237, “This behavior indicates that the order of the hydrophilic moieties in the amphiphilic ionic liquids decreases with increasing concentration of C12EO6. This change occurs because the EO chain of the surfactant is larger than the ammonium group of the amphiphilic ionic liquid, and the layer is formed by bending the EO chain.”
Comment:
(4) The conclusion reflects an overall summary of the field with further extension and includes future prospective – I would suggest clarifying this section.
Response:
- Thank you for your comments. We have removed the sentences “The surface tension of the amphiphilic gemini ionic liquids was not significantly affected by the spacer structure. However, ” on p.9, lines 342−343.
As mentioned above for Reviewer 1, we have added the sentences “The visualization using cryo-TEM will be useful for investigating nanostructure of ionic liquids in the future.” on p.9, lines 353−354. Further, we have added the sentences “These bulk properties can also be used for solubilization of various compounds as well as surfactants in ionic liquids.” on p.10, 360−361.
Comment:
(5) Authors should work on editing text, spacing pause, Justify, fonts, italics, etc.
Response:
- Thank you for pointing that out. We have written this manuscript using a template file of Molecules. We have added some revision to manuscript.
We have amended “ammmonium” to “ammonium” on p.1, line 29.
We have added “-” and edited the spacing between characters on p.8, lines 300 and 302.
We have changed “( )” to “[ ]” on p.2, lines 46−48 and 79−80.
Comment:
(6) References should be adapted to the requirements of the journal.
Response:
- Thank you for pointing that out. We have confirmed the format of references of Molecules, however, there was no noticeable problem. We have removed “.” of the reference 9 on p.10, line 402.
Comment:
(7) The style and grammar leave much to be desired in many places, some parts of the text are difficult to understand.
Response:
- We regret that there were problems with the English in our paper. We have amended manuscript as mentioned above. Furthermore, the manuscript has been carefully revised by a professional language editing service to improve grammar and readability.
Others
- We have amended “M.K.” to “M.N.” in Author Contribution on p.10, line 372.

Round 2
Reviewer 2 Report
I would like to support this revised paper (Manuscript number molecules-968493-peer-review-v2) for the publication in Molecules. All suggested changes were made (or discussed/clarified) by the authors. The results are informative, and discussion is clear. Summarizing, I think that this paper can be published as is.